# Computed Tomography Effective Dose and Image Quality in Deep Learning Image Reconstruction in Intensive Care Patients Compared to Iterative Algorithms

Emilio Quaia *, Elena Kiyomi Lanza de Cristoforis, Elena Agostini and Chiara Zanon

Department of Radiology, University of Padova, Via Giustiniani 2, 35128 Padova, Italy
* Correspondence: emilio.quaia@unipd.it; Tel./Fax: +39-(049)-821-2375

**Abstract:** Deep learning image reconstruction (DLIR) algorithms employ convolutional neural networks (CNNs) for CT image reconstruction to produce CT images with a very low noise level, even at a low radiation dose. The aim of this study was to assess whether the DLIR algorithm reduces the CT effective dose (ED) and improves CT image quality in comparison with filtered back projection (FBP) and iterative reconstruction (IR) algorithms in intensive care unit (ICU) patients. We identified all consecutive patients referred to the ICU of a single hospital who underwent at least two consecutive chest and/or abdominal contrast-enhanced CT scans within a time period of 30 days using DLIR and subsequently the FBP or IR algorithm (Advanced Modeled Iterative Reconstruction [ADMIRE] model-based algorithm or Adaptive Iterative Dose Reduction 3D [AIDR 3D] hybrid algorithm) for CT image reconstruction. The radiation ED, noise level, and signal-to-noise ratio (SNR) were compared between the different CT scanners. The non-parametric Wilcoxon test was used for statistical comparison. Statistical significance was set at $p < 0.05$. A total of 83 patients (mean age, $59 \pm 15$ years [standard deviation]; 56 men) were included. DLIR vs. FBP reduced the ED ($18.45 \pm 13.16$ mSv vs. $22.06 \pm 9.55$ mSv, $p < 0.05$), while DLIR vs. FBP and vs. ADMIRE and AIDR 3D IR algorithms reduced image noise ($8.45 \pm 3.24$ vs. $14.85 \pm 2.73$ vs. $14.77 \pm 32.77$ and $11.17 \pm 32.77$, $p < 0.05$) and increased the SNR ($11.53 \pm 9.28$ vs. $3.99 \pm 1.23$ vs. $5.84 \pm 2.74$ and $3.58 \pm 2.74$, $p < 0.05$). CT scanners employing DLIR improved the SNR compared to CT scanners using FBP or IR algorithms in ICU patients despite maintaining a reduced ED.

**Keywords:** CT; intensive care; reconstruction; algorithms; deep learning

## 1. Introduction

CT image reconstruction has evolved from the original filtered back projection (FBP) to hybrid and model-based iterative reconstruction (IR) algorithms, with a significant decrease in the radiation dose [1]. The main advantage of FBP is its computational efficiency, whereas its disadvantages include significant noise at low radiation doses and limited artifact reduction [1]. Iterative reconstruction (IR) algorithms are widely employed in CT image reconstruction to preserve image quality, even in low-dose CT acquisitions, with reduced image noise and artifacts [1,2]. Hybrid IR algorithms employ both FBP and IR algorithms (ranging from 50% to 90% with complementary levels of FBP) and allow fast CT image reconstruction with a reduction in image noise [1] and an improvement in image quality at lower radiation doses. Model-based IR algorithms are fully IR algorithms that use forward and backward reconstruction steps from the sinogram domain to the image domain [1]. The main advantage of model-based IR is the maintenance of CT image quality with low noise, even at low doses; however, its disadvantage is the need for high computational power and low capability in the detection rate of low-contrast structures on low-dose CT images [1–4].

In recent years, there has been growing interest in the application of deep learning image reconstruction (DLIR) algorithms. DLIR is a recently introduced CT image recon-

struction algorithm based on deep learning which employs convolutional neural networks (CNNs) for CT image reconstruction to produce CT images with a very low noise level, even at low radiation effective dose (ED) [4]. CNNs handle millions of parameters trained with thousands of paired high-quality, high-ED, and low-noise ground-truth CT images obtained from a large number of phantoms and patients. After training, a low-dose sinogram is provided to the CNN, and a final image with a very low noise level is obtained by comparing the output image to a ground-truth image across multiple parameters such as image noise, low contrast resolution, low contrast detectability, and noise texture. The backpropagation operation reports the differences to the network, which then strengthens some equations and weakens others, and the process is repeated until there is a proximity between the output and ground-truth images. The performance of DLIR algorithms for CT image reconstruction relies mainly on the quality and quantity of the training data and high-quality reference ground-truth CT images [5,6]. Commercially available DLIR algorithms include direct algorithms that use ground-truth images reconstructed by FBP and sinogram data directly fed into a CNN–True Fidelity (GE Healthcare) and Precise Image (Philips Healthcare) and indirect algorithms that use ground-truth images reconstructed by model-based IR algorithms (AiCE, Canon Medical System). The DLIR strengths of the FBP and IR algorithms can be selected by the operator as low (DLIR-L), medium (DLIR-M), or high (DLIR-H). In our study, we employed high DLIR strength according to the default settings of the CT equipment.

A marked reduction in radiation ED is particularly required in ICU patients who are exposed to high radiation exposure, frequently higher than 100 mSv during a single hospital admission, and particularly in those patients with prolonged hospitalization time [7,8] due to the extremely frequent use of X-ray imaging modalities. In particular, ICU patients undergo frequent CT scans, often with extended scanning lengths, hampered by low image quality and artifacts due to external or internal medical devices and patient arms placed along the body. No previous study has provided an intra-patient comparison between IR/FBP and DLIR CT reconstruction algorithms implemented in different CT scanners in terms of the radiation ED and image quality in ICU patients.

The aim of the present study was to assess whether the DLIR algorithm may reduce the CT effective dose (ED) and improve CT image quality in comparison with FBP and iterative reconstruction (IR) algorithms in ICU patients.

## 2. Materials and Methods

### 2.1. Patients

This study was conducted in accordance with the Declaration of Helsinki and approved by the Ethics Committee of our hospital (Prot. n. 0000569 approved on 4 January 2023). Patient informed consent was waived due to the retrospective nature of this study. We initially identified all consecutive patients referred to the ICU of our hospital because of their severe clinical status, major traumas, or even recent thoracic or abdominal major surgery (extended tumor resection or liver, cardiac, or lung transplant) between 1 October 2021 and 28 February 2023. Subsequently, we retrospectively selected only those patients who underwent at least two subsequent chest and/or abdominal contrast-enhanced CT scans with comparable scan lengths covering the same body region (chest, abdomen, or both chest and abdomen) during the same hospital admission. The DLIR algorithm was used in the first CT scan for CT image reconstruction, whereas FBP or Adaptive Iterative Dose Reduction 3D (AIDR3D) hybrid or even Advanced Modeled Iterative Reconstruction (ADMIRE) model-based IR algorithms were used in the second CT scan. To ensure that significant physical changes in patient features, including body mass index, occurred between the two CT scans, only those obtained within a limited timeframe of 30 days were included.

### 2.2. CT Scanning Protocols

Because it was not possible to use different CT image reconstruction algorithms on the same raw data obtained from the same patients owing to the different CT acquisition tech-

nical settings related to the subsequent CT image reconstruction algorithm (DLIR) applied to CT scans acquired with a lower tube kV and mAs than IR or FBP, we compared different CT scanners equipped with different CT image reconstruction algorithms, according to the manufacturer's technical solutions (Table 1). CT images were reconstructed at 3 mm and with a 512 × 512-pixel matrix. The DLIR strength was set to the highest level (DLIR-H), according to the manufacturer's default settings.

**Table 1.** CT scanning parameters. FBP = filtered back projection; DLIR = deep learning image reconstruction; ADMIRE = Advanced Modeled Iterative Reconstruction model-based iterative reconstruction algorithm; AIDR 3D = Adaptive Iterative Dose Reduction 3D—hybrid iterative reconstruction algorithm.

| System (Vendor) | Reconstruction Algorithm | Pitch |
|---|---|---|
| Somatom Sensation 64 (Siemens Healthineers, Enlargen, Germany) | FBP | 0.8 |
| Somatom Definition Edge (Siemens Healthineers, Enlargen, Germany) | ADMIRE | 0.6 |
| Aquilion ONE (Canon Medical Systems, Otawara-shi, Tochigi, Japan) | AIDR 3D | 0.81 |
| Revolution Evo (GE Healthcare, Chicago, IL, USA) | DLIR | 0.51 |

In every patient, CT was performed craniocaudally with a scan range from the lower neck to the costophrenic angle level on chest CT and from the diaphragm level to the pelvis on abdominal CT before and after iodinated contrast agent injection (ioexol 350 mgI/mL; Omnipaque 350, GE HealthCare, Barrington, IL, USA) or iodixanol 270 mg/mL (Visipaque 270, GE Healthcare, Barrington, IL, USA), iopromide 370 mgI/mL (Ultravist 370, Bayer, Leverkusen, Germany), or iomeprol 400 mgI/mL (Iomeron 400, Bracco, Milan, Italy). Patients were scanned with their arms placed along the body owing to their critical clinical status. The volume of contrast medium was calculated based on the patient's lean body weight (LBW) which was estimated from the patient's weight, height, and gender using Boer's equation [9]. The arterial phase was triggered by placing a region of interest (ROI) over the abdominal CT scan at the level of the second lumbar vertebral body and starting the scan when the density level achieved 100 HU. The portal venous and late phases were obtained at 70 and 180 s after iodinated contrast injection. The contrast agent was injected into the antecubital vein (total contrast volume and injection speed adjusted by the patient's body weight to 3–4 mL/s) and saline push (10 s at the same rate). The following CT parameters were used: tube voltage, 100–120 kVp; automatic tube current modulation; gantry rotation period, 280 ms; and detector collimation, 0.625 mm. The CT dataset was then reconstructed at 1.25 mm section thicknesses with 512 × 512 matrices, using standard kernels for soft tissues.

Although the same scanning protocol was generally used in both the first and second CT scans, a mismatch in scanning length, presence or absence of unenhanced CT scans, or even the number of contrast-enhanced dynamic phases (arterial, portal venous, or delayed phases) was possible between the two subsequent thoracic and/or abdominal CT scans. Therefore, these patients were excluded from analysis. In patients who underwent more than two repeated CT scans, only the two closest CT scans reconstructed using the DLIR and FBP or IR algorithms were considered for analysis.

Generally, the same iodinated contrast agent dosage and concentration were used in both the first and second CT scans unless the use of a different iodinated contrast agent type was required (e.g., suspicion of bleeding after major surgery, change in iodinated contrast type and/or injected contrast volume due to anaphylactoid reaction or incoming acute kidney injury, even suspicious pulmonary embolism). Patients in whom the iodinated contrast agent type, injected volume, and/or iodine dose was changed or modified were excluded from the analysis.

## 2.3. Radiation Effective Dose Analysis

The CT dose index volume (CTDIvol) and dose-length product (DLP) were obtained retrospectively from CT dose reporting produced automatically by the CT equipment at the end of the scan and archived on the PACS. The radiation ED was calculated by multiplying the DLP by the body-region-specific conversion coefficient, k, according to the ICRP recommendations [10,11].

## 2.4. Visual Image Quality Analysis

Visual image quantitative analysis was performed 2 weeks before quantitative analysis. Two radiologists with 3 and 10 years of experience performed independent subjective analyses of the three groups of images. The radiologists were blinded to the image reconstruction techniques and patient characteristics. The images were displayed in random order in a preset window, displaying a sequence at a time. The radiologists were able to scroll through the images and adjust the window width and position randomly. We used a 5-point scale, according to Table 2.

**Table 2.** Subjective visual image quality of mediastinal and abdominal parenchyma tissues.

| Score | Definition |
|---|---|
| 1 | Poor definition of parenchyma borders and clearly visible noise (unacceptable image) |
| 2 | Moderate definition of parenchyma borders and moderately visible noise (suboptimal image) |
| 3 | Moderate definition of parenchyma borders and barely visible noise (acceptable image) |
| 4 | Good definition of parenchyma borders and barely visible noise (good image) |
| 5 | Excellent definition of parenchyma borders and very low image noise (optimal image) |

## 2.5. Quantitative Image Quality Analysis

CT image noise was calculated off-site on a dedicated PC using MATLAB (MATLAB version: 9.13.0 (R2022b), Natick, MA, USA: The MathWorks Inc.; 2022) with the Global Noise Level (GNL) algorithm for automatic noise measurement [12,13] by a medical student with specific competence in CT image quantitation software analysis over 4 years. The GN algorithm was used to analyze only the selected slice images. Observers selected similar slice locations, and therefore approximately similar noise, by using anatomical landmarks for slice selection. This assumption was tested by measuring the variation in the selected slice locations across the observers. The slice locations selected by the observers were averaged, and the slice image closest to this location was selected for GN analysis.

To objectively compare image quality, the signal-to-noise ratio (SNR) was measured for different reconstruction algorithms (FBP, DLIR, AIDR 3D, and ADMIRE). For thoracic evaluation, the SD of the values in Hounsfield units (HUs) was measured in regions of interest (ROIs) measuring $\geq 1\ cm^2$ drawn in the bilateral abdominal fat (SDax1 and SDax2), and the average HU values were measured in the bilateral paraspinal muscles (HUPSM1 and HUPSM2). The noise and SNR for each scan were calculated using the following equations:

$$\text{Noise} = (\text{SDax1} + \text{SDax2})/2$$

$$\text{SNR} = (\text{HUPSM1} + \text{HUPSM2})/(\text{SDax1} + \text{SDax2})$$

## 2.6. Statistical Data Analyses

Statistical data analyses were performed using SciPy 1.11.2, an open-source software using Python 3.12 programming language, by a medical student with specific competence in statistical software analysis over 4 years. After the Shapiro–Wilk test failed to show a normal distribution, the Wilcoxon signed-rank test for paired data was used to assess the differences between the FBP, IR, and DLIR effective doses and image quality. The minimal appropriate patient number ($n$ = 65) to be included in this study was estimated by considering a statistical power of 0.8, a significance criterion of 0.05, and a standard deviation of 20 for the effective dose expressed in mSv. Cohen's kappa statistic was

calculated for an agreement on the independent scoring of the image quality between the two radiologists. A kappa statistic of 0.81~1.00 implies an excellent agreement; 0.61~0.80, a substantial agreement; 0.41~0.60, a moderate agreement; 0.21~0.40, a fair agreement; and 0.00~0.20, a poor agreement. For all statistical tests, a *p* value < 0.05 was set to indicate a statistically significant difference.

## 3. Results

### 3.1. Patients

Figure 1 shows the patient flow chart.

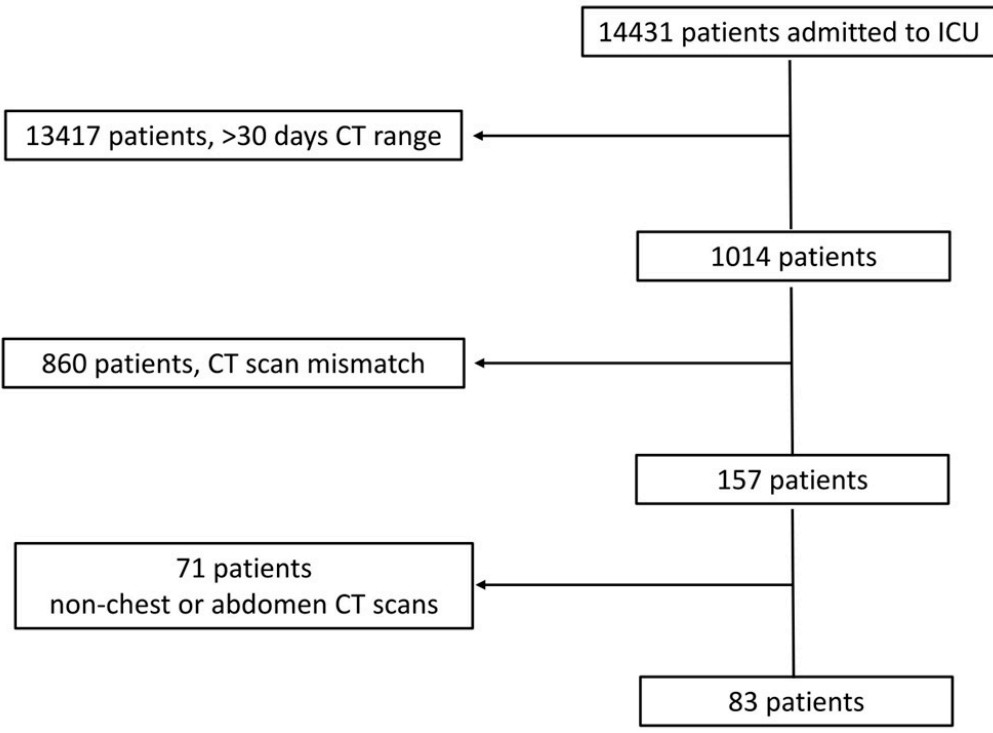

**Figure 1.** Patient flow chart.

Initially, we identified 14,431 patients who were admitted to the ICU. We excluded 13,417 patients due to a temporal distance of >30 days between the two subsequent CT scans; 860 patients due to a mismatch in the scanning length between the two CT scans (*n* = 350) or differences in CT scanning protocols (*n* = 251), including the absence of an unenhanced scan, different numbers of dynamic phases, or even changes in contrast agent type and/or contrast volume administration (*n* = 259); and 71 patients due to CT scans that did not include the chest and/or abdomen (e.g., brain and limb). The total hospitalization period was 10–45 days (mean ± SD, 22 ± 10 days).

Finally, we included 83 patients (Table 3) who underwent CT scans of the chest (*n* = 14; 5 patients were scanned on unenhanced CT and during the arterial phase, while 9 patients underwent both unenhanced CT and contrast-enhanced CT on arterial and portal venous phases), abdomen (*n* = 51; 32 patients scanned on unenhanced CT and arterial phase and 19 patients scanned both on unenhanced CT and arterial and portal venous phases), and both chest and abdomen (*n* = 18; 12 patients scanned on unenhanced CT and arterial phases and 6 patients scanned both on unenhanced CT and arterial and portal venous phases). The timeframe between the two CT scans considered for quantitative analysis was 10.8 ± 8.6 days (range, 1–30 days).

### 3.2. Visual Analysis

The visual analysis results showed significant differences in the image quality of soft tissue among the three reconstruction methods (all *p* < 0.05) (Table 4). The image score of DLIR (mean score = 5) was higher than that of ADMIRE (mean score = 4), AIDR 3D (mean score = 3), and FBP (mean score = 3) (Table 4). Both radiologists believed that DLIR had outstanding noise reduction. The subjective scores of the two radiologists were consistent (kappa value range: 0.48–0.91) (Figure 2).

**Table 3.** Patient features.

| Patients | Total | FBP vs. DLIR | ADMIRE vs. DLIR | AIDR 3D vs. DLIR |
|---|---|---|---|---|
| Patients included | 83 | 12 | 59 | 12 |
| Male/female | 56/27 | 9/3 | 37/22 | 10/2 |
| Age, years mean $\pm$ SD (range) | 59 $\pm$ 15 (31–73) | 64 $\pm$ 8 (52–64) | 54 $\pm$ 16 (34–73) | 50 $\pm$ 27 (31–70) |
| CT time interval, days mean $\pm$ SD (range) | 11 $\pm$ 9 (1–30) | 17 $\pm$ 10 (2–29) | 9 $\pm$ 8 (1–30) | 11 $\pm$ 8 (2–27) |
| **CT scans** | | | | |
| Chest CT (percentage) | 14/83 (17%) | 4/12 (33%) | 10/59 (17%) | 0/12 (0%) |
| Abdomen CT (percentage) | 51/83 (61%) | 5/12 (42%) | 35/59 (59%) | 11/12 (92%) |
| Chest and abdomen CT (percentage) | 18/83 (22%) | 3/12 (25%) | 14/59 (24%) | 1/12 (8%) |

**Table 4.** Dose analysis results. * indicates a *p* value < 0.05, compared with DLIR. FBP = filtered back projection; ADMIRE = Advanced Modeled Iterative Reconstruction model-based iterative reconstruction algorithm; AIDR 3D = Adaptive Iterative Dose Reduction 3D—hybrid iterative reconstruction algorithm; DLIR = deep learning image reconstruction; CTDIvol = CT dose index volume; DLP = dose-length product. DLIR differed significantly only from FBP in terms of radiation dose, while the difference was not found to be significant between DLIR and iterative algorithms and between FBP and iterative algorithms ADMIRE and AIDR 3D.

| Algorithm | CTDI (mGy) Mean $\pm$ SD, Range | DLP (mGy $\times$ cm) Mean $\pm$ SD, Range | Effective Dose (mSv) Mean $\pm$ SD, Range |
|---|---|---|---|
| FBP | 29.5 $\pm$ 12.46 (6.41–51.79) * | 1476.81 $\pm$ 626.30 (284–2105) * | 22.06 $\pm$ 9.55 (3.98–31.57) * |
| ADMIRE | 29.42 $\pm$ 5.86 (3.52–35.27) | 1472.86 $\pm$ 1191.68 (106–6778) | 22.19 $\pm$ 17.91 (1.48–101.67) |
| AIDR 3D | 30.83 $\pm$ 5.86 (3.52–35.27) | 1545.35 $\pm$ 1191.68 (106–6778) | 23.08 $\pm$ 17.91 (1.48–101.67) |
| DLIR | 24.67 $\pm$ 61.01 (3.45–355.42) | 1235.53 $\pm$ 873.67 (145.89–4528.42) | 18.45 $\pm$ 13.16 (2.04–67.93) |

### 3.3. Effective Dose and Quantitative Analysis

DLIR differed significantly only from FBP in terms of radiation dose, while the difference was not found to be significant between DLIR and iterative algorithms and between FBP and iterative algorithms ADMIRE and AIDR 3D (Table 4). DLIR improved both the image noise and SNR compared to both the FBP and IR algorithms (Table 5). Among IR algorithms, compared to AIDR 3D, ADMIRE provided similar exposure data (Table 4) with lower noise and a higher SNR (Table 5).

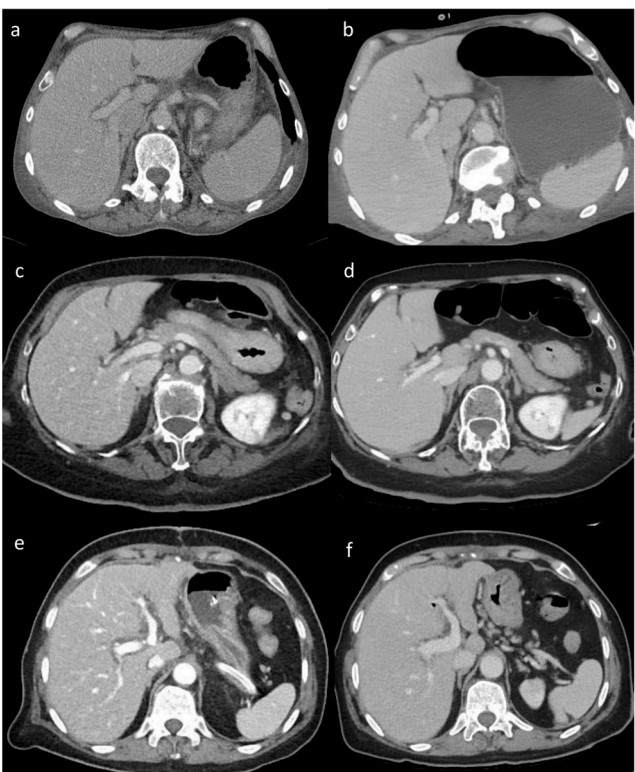

**Figure 2.** (**a**–**f**) Visual analysis. Visual differences in abdominal parenchyma border definition and noise among different reconstruction algorithms: (**a**,**b**) 45-year-old male patient after major surgery; (**c**,**d**) 47-year-old male patient after lung transplant; and (**e**,**f**) 55-year-old male patient after cardiac transplant. Filtered back projection (FBP) (**a**), Advanced Modeled Iterative Reconstruction (ADMIRE) model-based iterative reconstruction (**c**), and Adaptive Iterative Dose Reduction 3D (AIDR 3D) hybrid iterative reconstruction (**e**) vs. deep learning image reconstruction (DLIR) (**b**,**d**,**f**). FBP, ADMIRE, and AIDR 3D were scored as 2, 4, and 3, respectively, whereas DLIR images were scored as 5 by all reviewers.

**Table 5.** Image quality results. * indicates a *p* value < 0.05 compared to DLIR. FBP = filtered back projection; ADMIRE = Advanced Modeled Iterative Reconstruction model-based iterative reconstruction algorithm; AIDR 3D = Adaptive Iterative Dose Reduction 3D—hybrid iterative reconstruction algorithm; DLIR = deep learning image reconstruction. DLIR improved the visual score, image noise, and SNR compared to both the FBP and IR algorithms.

| Algorithm | Visual Analysis Mean (Range) | Noise HU Mean ± SD (Range) | SNR HU Mean ± SD (Range) |
|---|---|---|---|
| FBP | 3 (1–3) * | 14.85 ± 2.73 (11.50–18.94) * | 3.99 ± 1.23 (2.37–6.15) * |
| ADMIRE | 4 (2–4) * | 14.77 ± 32.77 (7.33–105.50) * | 5.84 ± 2.74 (0.21–8.71) * |
| AIDR 3D | 3 (2–3) * | 11.17 ± 32.77 (7.33–105.50) * | 3.58 ± 2.74 (0.21–8.71) * |
| DLIR | 5 (4–5) | 8.45 ± 3.24 (4.29–18.19) | 11.53 ± 9.28 (6.55–30.30) |

## 4. Discussion

Radiation ED in CT is determined by technical parameters (kV, mA, collimation, and pitch) employed in the acquisition phase. Reconstruction algorithms do not directly reduce the radiation dose but may compensate for image quality loss due to a reduction in radiation dose or may improve the image quality by maintaining a constant radiation dose. In our study, we found that DLIR improved the SNR compared to both the FBP and IR algorithms in ICU patients despite maintaining a reduced ED.

ICU patients are generally exposed to high radiation doses due to frequent and extended chest and/or abdominal CT scans, especially in patients who undergo major

surgery or organ transplant and in patients with prolonged hospitalization time, as in the ICU patients included in our study. Therefore, we focused on the ICU patient cohort because our aim was to analyze the major advantages of DLIR in terms of radiation dose exposure and CT image quality under extreme clinical conditions, which justified the use of repeated CT scans over a relatively restricted period.

Our study confirmed that DLIR for CT images provides significant benefits in terms of image quality over the FBP and IR algorithms despite maintaining a reduced ED in ICU patients, which emphasizes the advantage of the DLIR approach and its potential in daily clinical practice, in keeping with previously published papers [14–16].

In our study, DLIR provided a reduction in radiation ED compared to both IR algorithms included in our study, although this result did not achieve statistical significance. This was due to the selected CT technical acquisition factors, including the tube voltage and automatic tube current modulation grade, which were similar between the different CT scanners. Consequently, the radiation ED did not change significantly, with the advantage of reduced CT image noise, owing to the use of the DLIR algorithm. Most likely, DLIR may provide a reduction in the radiation dose, even when compared to IR algorithms, provided that a similar image quality in terms of both the noise level and SNR between the DLIR and IR algorithms is preliminarily planned. In this case, a comparable SNR between CT images produced by CT scanners employing DLIR vs. those scanners employing IR algorithms would imply a higher patient radiation ED in CT scanners using IR, which is related to the higher tube current required to reduce noise. This reflects the generally higher attention paid to CT image quality than to patient radiation exposure in general clinical practice, even if repeated CT scans are required over a limited temporal range to strictly monitor clinical evolution, as in ICU patients.

DLIR may provide an improved image quality even with a reduced ED in ICU patients who are frequently examined using different X-ray imaging modalities, including plain X-ray film and CT scans. CT scans provide the highest dose from medical exposure, although they are often penalized by low image quality in ICU critical patients. The main result is that all the advantages provided by DLIR algorithms translate into safer imaging practices, higher diagnostic confidence and more accurate diagnosis from radiologists, and, ultimately, better patient care. Considering this increase, it is reasonable to expect a wider implementation of DLIR algorithms in the future given the increasing computational power of CT scanners. However, further evaluation is needed to investigate the potential differences between DLIR and IR algorithms, even in other anatomical locations such as the head or limbs, or in specific diseases, and to assess whether the improved image quality provided by DLIR may significantly affect subjective CT image quality, CT workflow, and efficiency in terms of the time needed to assess CT images or to achieve the correct diagnosis by a radiologist.

The strict inclusion criteria employed in this study to minimize the intrinsic bias related to the different CT scanners employed determined a reduced sample size with a consequently wide standard deviation. However, the minimal appropriate patient number ($n = 65$), which is lower than the number of patients finally included in this study, was estimated by considering a statistical power of 0.8, a significance criterion of 0.05, and a standard deviation of 20 for the effective dose expressed in mSv.

The first limitation of the present study includes the approximate approach we used for estimating the radiation ED based on the DLP obtained by multiplying the CTDIvol by the scan length by the body-region-specific conversion coefficient k [10,11]. Monte Carlo (MC) simulation is generally considered the most accurate method for estimating radiation ED, owing to its ability to provide an effective and realistic model of the physical interactions between radiation and tissues, considering the CT source, filtration, tube current, and scanner geometry [10,11]. Moreover, a size-specific dose estimate (SSDE) would have been useful in patient dose comparison, provided that patient lateral and anteroposterior diameters are known, although the SSDE does not take the organs in the CT scan's field

of view into account and is not a measure of ED. The SSDE is a better estimate of patient radiation dose from CT than CTDIvol in systems that use automated exposure control [17].

The second limitation of this study corresponds to the wide variation in CT scanner characteristics and technical features related to the use of different CT image reconstruction algorithms, detector designs and configurations, dose modulation algorithms, and patient positioning/handling which may affect the outcomes of this study. However, there is no other way to compare the CT image quality from the same patient by using different CT scanners employing different reconstruction algorithms. The basic CT image technical parameters—tube voltage, automatic tube current modulation, detector collimation, section thicknesses, image matrix, kernels, iodinated contrast agent type, injected volume, and/or iodine dose—were kept constant between different scanners. A mismatch in scanning length, presence or absence of unenhanced CT scans, or even in the number of contrast-enhanced dynamic phases (arterial, portal venous, or delayed phases) determined the exclusion of patients from the analysis.

Other limitations are the retrospective nature of this study, the reduced sample size, and the wide patient population with very different clinical features.

**5. Conclusions**

In conclusion, CT scanners employing DLIR improved the SNR compared to CT scanners using FBP or IR algorithms in ICU patients despite maintaining a reduced ED.

**Author Contributions:** Conceptualization, E.Q.; methodology, E.Q.; software, E.K.L.d.C.; validation, E.Q., E.K.L.d.C., E.A. and C.Z.; formal analysis, E.K.L.d.C.; investigation, E.Q., E.K.L.d.C., E.A. and C.Z.; resources, E.Q. and E.K.L.d.C.; data curation, E.K.L.d.C.; writing—original draft preparation, E.Q.; writing—review and editing, E.Q. and C.Z.; visualization, E.K.L.d.C.; supervision, E.Q.; project administration, E.Q. All authors have read and agreed to the published version of the manuscript.

**Funding:** This research received no external funding.

**Institutional Review Board Statement:** This study was conducted in accordance with the Declaration of Helsinki and approved by the Ethics Committee of our hospital (Prot. n. 0000569 approved on 4 January 2023).

**Informed Consent Statement:** Patient informed consent was waived due to the retrospective nature of this study.

**Data Availability Statement:** Data are not publicly available due to privacy or ethical restrictions, although they can be provided after anonymization.

**Conflicts of Interest:** The authors declare no conflicts of interest.

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
