# Peer review of "Computed Tomography Effective Dose and Image Quality in Deep Learning Image Reconstruction in Intensive Care Patients Compared to Iterative Algorithms"

_tomography, doi:10.3390/tomography10060069_

Round 1

Reviewer 1 Report

Comments and Suggestions for Authors

1. The paper employs a well-established DLIR approach using convolutional neural networks for CT image reconstruction. However, the methodology itself does not appear to offer any significant methodological innovation. The authors should clarify how their approach differs from and improves upon previous work in this field, especially for the initensive care patients.

2. The use of two-stage CT scan data from the same patients, with potential differences in scanner manufacturers between the scans, introduces significant confounders into the study. Even with data screening, the differences in scanner technology and calibration could bias the results. The reported reduction in ED with DLIR compared to FBP is statistically significant but may not be clinically meaningful given the potential biases in the data. The finding should be interpreted with caution due to the potential for scanner-related biases.

Author Response

Rev 1, comment 1: The paper employs a well-established DLIR approach using convolutional neural networks for CT image reconstruction. However, the methodology itself does not appear to offer any significant methodological innovation. The authors should clarify how their approach differs from and improves upon previous work in this field, especially for the intensive care patients.

The authors understand the reviewer concern. Previous literature addressed the difference in scanning protocols, image quality, and dose of exposition between different CT vendors and algorithms by using phantoms. No previous study has provided an intra-patient comparison between FBP, iterative algorithms (hybrid or even model-based), and DLIR CT image reconstruction algorithm implemented in different CT scanners in terms of the radiation dose and image quality in Intensive care patients. Please see text for changes.

Rev 1, comment 2: The use of two-stage CT scan data from the same patients, with potential differences in scanner manufacturers between the scans, introduces significant confounders into the study. Even with data screening, the differences in scanner technology and calibration could bias the results. The reported reduction in ED with DLIR compared to FBP is statistically significant but may not be clinically meaningful given the potential biases in the data. The finding should be interpreted with caution due to the potential for scanner-related biases.

The authors understand the reviewer concern which is addressed in the study limitation section. This actually represents the main limitation of this study due to the wide variation in CT scanner technical features related to the use of different CT image reconstruction algorithms, detector designs and configurations, dose modulation algorithms, and patient positioning/handling. However, there is no other way to compare the CT image quality from the same patient by using different CT scanners employing different reconstruction algorithms. In this study the basic CT image technical parameters - tube voltage, automatic tube current modulation, detector collimation, section thicknesses, image matrix, and kernels, iodinated contrast agent type, injected volume, and/or iodine dose - were kept constant between different CT scanners. A mismatch in scanning length, presence or absence of unenhanced CT scans, or even the number of contrast-enhanced dynamic phases (arterial, portal venous, or delayed phases) determined the exclusion of patients from the subsequent analysis (please, see text for changes). 

Reviewer 2 Report

Comments and Suggestions for Authors

This paper evaluates deep learning image reconstruction (DLIR) algorithms for CT image reconstruction, against filtered back projection (FBP) and iterative reconstruction (IR) algorithms for noise reduction and image quality. In particular, the aim is to assess if DLIR reduces the CT effective dose (ED) required. It was found that CT scanners using DLIR improved SNR compared to scanners using FBP or IR, and also reduced ED for scanners using FBP.

Some issues might be considered:

1. In Section 2.5, subjective image quality ranging from 1 to 5 is described. Is this image quality metric analysed later on, sinec Table 4 does not appear to reference it?

2. For Section 2.6 and Table 4, the justification for the methodology using GNL might be further clarified. In particular, the Discussion section states that ground truth (i.e. noiseless) images are used for model development. Then, were the noise and SNR metrics compared against noiseless reference ground truth here, and if so, how was the (assumed) noiseless ground truth obtained? If not, then would the noise and SNR metrics be meaningful, since it is trivially possible to reduce noise (e.g. by blurring the image) while degrading important image details?

3. In Section 3.3, the estimation of effective dose (ED) might be clarified further, since it appears that the actual radiation dose is determined prior to the actual CT scan. In particular, how the actual radiation dose is determined (e.g. is it standard for each CT scanner/algorithm, or might it be adjusted depending on patient demographics) might be explained.

4. For Table 3, the analyses might be rechecked, since only the FBP results are claimed to be significant at P<0.05, whereas the results for ADMIRE are all not significant at that level, despite the means being very similar to FBP, and the SD both being smaller and larger for the different algorithms.

Author Response

Rev 2, comment 1: 1. In Section 2.5, subjective image quality ranging from 1 to 5 is described. Is this image quality metric analysed later on, since Table 4 does not appear to reference it?

Visual image quantitative analysis was performed 2 weeks before quantitative analysis. The results of visual analysis are now included in Table 4 according to the reviewer suggestions (please see text for changes).

Rev 2, comment 2: For Section 2.6 and Table 4, the justification for the methodology using GNL might be further clarified. In particular, the Discussion section states that ground truth (i.e. noiseless) images are used for model development. Then, were the noise and SNR metrics compared against noiseless reference ground truth here, and if so, how was the (assumed) noiseless ground truth obtained? If not, then would the noise and SNR metrics be meaningful, since it is trivially possible to reduce noise (e.g. by blurring the image) while degrading important image details?

Ground truth training data are CT images reconstructed by FBP that can faithfully represent the scanned object/patient. Low-noise ground-truth CT images are used to train the CNN for low-dose CT image reconstruction and are not available in the scanner since they are property of the manufacturer. High quality reference ground-truth CT images are stored in a separate cloud just for CNN reconstruction algorithm. Consequently, no ground truth image were available for quantitative image analysis (please see text for changes).

Rev 2, comment 3:  In Section 3.3, the estimation of effective dose (ED) might be clarified further, since it appears that the actual radiation dose is determined prior to the actual CT scan. In particular, how the actual radiation dose is determined (e.g. is it standard for each CT scanner/algorithm, or might it be adjusted depending on patient demographics) might be explained.

The CT dose index volume (CTDIvol) and dose-length product (DLP) were obtained retrospectively from CT dose reporting produced automatically by the CT equipment at the end of the scan and archived on the PACS (please, see text for changes). The radiation effective dose in mSv was calculated by multiplying the DLP by body region– specific conversion coefficient, k, according to the ICRP recommendations.

Rev 2, comment 4:  For Table 3, the analyses might be rechecked, since only the FBP results are claimed to be significant at P<0.05, whereas the results for ADMIRE are all not significant at that level, despite the means being very similar to FBP, and the SD both being smaller and larger for the different algorithms.

As it is now reported on Table 3 caption DLIR differed significantly only from FBP in terms of radiation dose while difference was found not significant between DLIR and iterative ADMIRE and AIDR 3D algorithms. Please see text for changes.  

Reviewer 3 Report

Comments and Suggestions for Authors

This study devoted in assessing whether the Deep learning image reconstruction (DLIR) algorithm can reduce the CT effective dose (ED) and improve CT image quality against filtered back projection (FBP) and iterative reconstruction (IR) algorithms in ICU patients. The authors identified all consecutive patients referred to the ICU who underwent at least two consecutive chest and/or abdominal contrast-enhanced CT scans within a period of 30 days using DLIR and subsequently FBP or IR algorithm (Advanced Modeled Iterative Reconstruction model-based algorithm [ADMIRE], or Adaptive Iterative Dose Reduction 3D [AIDR 3D]) for CT image reconstruction.

The radiation effective dose (ED), noise level, and SNR were compared between the different CT scanners. The non-parametric Wilcoxon test was used for statistical comparison with significance at p < 0.05. DLIR vs FBP reduced ED (18.45 ± 13.16 mSv vs 22.06 ± 9.55 mSv, P < 0.05), while DLIR vs FBP and vs ADMIRE and AIDR 3D IR algorithms reduced image noise (8.45 ± 3.24 vs 14.85 ± 2.73 vs 14.77 ± 32.77 and 11.17 ± 32.77) and SNR (11.53 ± 9.28 vs 3.99 ± 1.23 vs 5.84 ± 2.74 and 3.58 ± 2.74, P < 0.05). The authors concluded that CT scanners employing DLIR can reduce radiation ED and improve SNR compared to CT scanners using FBP, whereas CT scanners using DLIR improved SNR compared to CT scanners using FBP or IR algorithms in ICU patients.

Comments:

     1)     The authors wrote that they from identified 14,431 patients excluded 13,417 patients due to a temporal distance, 860 patients due to a mismatch in the scanning length; and 71 patients due to CT scans that did not include the chest and/or abdomen (e.g., brain and limb). So, they finally investigated 83 patients. This sample size is low. This reviewer thinks that authors should discuss this fact and explain how from statistical point of view the results could be changed for larger sample size. Also, one can see that according to SD of the numerical results obtained for reduced ED and SNR strongly speaking do not guarantee security conclusion about better performance of DLIR algorithm against other algorithms used in comparison.

      2) This reviewer proposes in final part of this study (lines 301-317), where there are mentioned the limitations of this study, discussing possible propositions that can help in resolving these drawbacks.

Author Response

Rev 3, comment 1:  The authors wrote that they from identified 14,431 patients excluded 13,417 patients due to a temporal distance, 860 patients due to a mismatch in the scanning length; and 71 patients due to CT scans that did not include the chest and/or abdomen (e.g., brain and limb). So, they finally investigated 83 patients. This sample size is low. This reviewer thinks that authors should discuss this fact and explain how from statistical point of view the results could be changed for larger sample size. Also, one can see that according to SD of the numerical results obtained for reduced ED and SNR strongly speaking do not guarantee security conclusion about better performance of DLIR algorithm against other algorithms used in comparison.

The strict inclusion criteria employed in this study to minimize the intrinsic bias related to the different CT scanner employed, determined a reduced sample size with a consequently wide standard deviation. However, the minimal appropriate patient number (n = 65), which is lower to the number of patients finally included in this study, was estimated by considering a statistical power of 0.8, a significance criterion of 0.05, and a standard deviation of 20 for the effective dose expressed in mSv. Please see text for changes.

Rev 3, comment 2:  This reviewer proposes in final part of this study (lines 301-317), where there are mentioned the limitations of this study, discussing possible propositions that can help in resolving these drawbacks.

Please see text for changes. We tried to add some possible solutions to the main limitations of the present study.

Round 2

Reviewer 1 Report

Comments and Suggestions for Authors

The paper exhibits a high level of redundancy, with the repetition rate exceeding acceptable standards. It is crucial to reduce the duplication rate to below 15% to ensure the integrity and originality of the research.

Author Response

The authors changes text according to the reviewer suggestions. The paragraph titled “CT image reconstruction algorithms” has been removed since all algorithm features are already present on Table 1. Moreover, one Table has been added describing all visual analysis parameters and all the following Tables have been renumbered. The first paragraph of discussion is now embedded on introduction to avoid text redundancy. Please see text for changes.

Reviewer 2 Report

Comments and Suggestions for Authors

We thank the authors for addressing our previous comments.

Author Response

No particular comment. The reviewer stated that all concerns have been addressed.